# Gibbs Free Energy and Reaction Rate Acceleration in and on Microdroplets

**DOI:** 10.3390/e21111044

**Published:** 2019-10-26

**Authors:** Adrian F Tuck

**Affiliations:** Physics Department, Imperial College London, London SW7 2AZ, UK; a.tuck@imperial.ac.uk

**Keywords:** microdroplets, reaction rates, Gibbs free energy, scale invariant analysis of observations

## Abstract

Recent observations show that many reactions are accelerated in microdroplets compared to bulk liquid and gas media. This acceleration has been shown to feature Gibbs free energy changes, Δ*G*, that are negative and so reaction enabling, compared to the reaction in bulk fluid when it is positive and so reaction blocking. Here, we argue how these Δ*G* changes are relatable to the crowding enforced by microdroplets and to scale invariance. It is argued that turbulent flow is present in microdroplets, which span meso and macroscales. That enables scale invariant methods to arrive at chemical potentials for the substances involved. *G* and Δ*G* can be computed from the difference between the whole microdroplet and the bulk medium, and also for individual chemical species in both cases, including separately the microdroplet’s surface film and interior, provided sufficiently fine resolution is available in the observations. Such results can be compared with results computed by quantum statistical mechanics using molecular spectroscopic data. This proposed research strategy therefore offers a path to test its validity in comparing traditional equilibrium quantum statistical thermodynamic tests of microdroplets with those based on scale invariant analysis of both their 2D surface and 3D interior fluid flows.

## 1. Introduction

The observed acceleration of chemical reaction rates in microdroplets, and in many cases their transformation from positive to negative Gibbs free energy changes (Δ*G*), has implications for chemistry across many areas, including basic kinetics, aerosols in both the laboratory and atmosphere, prebiotic chemistry, synthetic routes and industrial processes [1,2,3,4,5,6,7,8]. Here, we discuss in more detail how Δ*G* is influenced by the processes determining the rate of a reaction when it is taking place in and on a microdroplet, as contrasted to its value in the bulk liquid or gas. The implications of this phenomenon for the chemical future have been emphasized [6,7]. A searching experimental examination has been made very recently [8]. The current paper embodies a research proposal which seeks to calculate Gibbs free energy (*G*) and Δ*G* for open, non-equilibrium systems—which most microdroplets are—by using the methods of scale invariance.

The determination of entropy and Gibbs energy for open, non-equilibrium systems by analysis of observed scale invariance has been proposed and exemplified [9]. Basic principles expounded were that they and other statistical thermodynamic variables were obtainable by the methods of generalized scale invariance, with attractive forces being responsible for the production of organization, while repulsive ones effectuated dissipation and hence the necessary entropy production. For scale invariance to be manifest, the same process or processes must be operative scale by scale over the whole range; it was argued to be the dilution of energy density. Scale invariance has been observed over a vast range of scales, from microscopic through mesoscopic to macroscopic, including the atmosphere and astronomical molecular clouds. Dilution of energy density will minimize *G* and hence enable whatever chemical and fluid mechanical processes will lead the microdroplet to a stationary state, or in a few cases to equilibrium as defined by molecular statistical thermodynamics. Scale invariance has also been observed in two-dimensional films, including biological membranes. Transfer of reactant and product molecules within and between the 2D outer layer and the 3D interior will be by turbulent fluid flow, and so scale invariant.

The observation of scale invariance in the occurrence of monomeric residues in biopolymers, including both proteins and nucleic acids, makes possible a comparison between the sum of the chemical potentials (the Gibbs free energy of individual substances) of the molecular population of the microdroplet and its exterior film by this scale invariant method, with the results from established methods of statistical mechanics and spectroscopy.

In the context of atmospheric chemistry, polymerization of atmospheric molecules, particularly hydrocarbons with polar groups, has been observed [10,11,12], with long-tailed mass spectra to high masses in evidence. A microdroplet with chemical reactions on its surface, significant levels of charge, or which is subject to an actinic photon flux, will not be at equilibrium.

## 2. Methods

In viewing a microdroplet reactor, the most obvious effect is that of the restricted volume available to the reactants, leading to the classical but vaguely defined concept of ‘crowding’. Considering a 1 micrometer (10^−6^ m) radius droplet, it will be capable of holding thousands of small molecules, typically of nanometer (10^−9^ m) radius. The encasing layer or film will be of different composition to the interior, depending upon the detailed characteristics of the molecules involved; the solvent involved is commonly but not always water. For the law of mass action to be applicable, all reactant molecules must have random access to the entire volume in the time scale concerned. Restriction to less than three dimensions will accelerate reactions. It has been shown in the case of stratospheric ozone that this is a relevant consideration [9]. For scale invariance to be diagnosed and applied to the estimation of entropy and *G*, there must be three orders of magnitude in scale, a condition met with nanometric resolution in the observations. This scaling range will be defined by the three orders of magnitude between the inner scale of a few mean free paths [9] and the outer scale defined by the microdroplet’s circumference, much as a great circle defines the outer scale for the atmosphere.

Applying the principle that attractive forces produce organization in the form of reaction, these forces will be more effective in microdroplets than in bulk because they will be acting over closer ranges. The entropic price is paid by repulsive forces effecting dissipation and will also be constrained by the microdroplet, which has a smaller volume than that in bulk. Thus, both these effects result in a favorable effect on Δ*G*. ‘Organization’ can be, for example, the production of oligomers or polymers from monomers [9,12,13], with the reaction products at the bottom of their potential well representing dilution of energy density. It should be emphasized that the interior of a microdroplet is unlikely to be of the same uniform molar composition as bulk liquid, because the effects of interaction with the encasing layer will propagate into the interior, with the result that composition will have a dynamical gradient along a radius from the center to the exterior. The transfer will be fluid mechanical rather than diffusive in the Einstein–Smolukowski sense.

In a homogeneous droplet, the Gibbs energy of the confining layer is regarded as the surface tension of the bulk liquid. In a heterogeneous microdroplet, the situation is more complicated. Even for small polar molecules in aqueous solution, the outer layer will not be the same composition as the interior [14]. Where amphiphiles and surfactants are concerned, they may form films, which can be two-dimensional gases, liquids or solids, depending upon surface pressure, with significant chemical consequences [15,16,17]. In the case of a microdroplet with a liquid or solid exterior film, it can provide a radically different reaction medium from either the interior or the bulk. Particularly important in this context is the ability to form an anhydrous environment of the condensation reactions that eliminate water to produce biopolymers, including peptides, nucleotides and phosphorylation reactions [18,19,20,21].

The scale of microdroplets is such that they fall into the category of systems ranging from mesoscopic to macroscopic, thus being capable of non-equilibrium statistical thermodynamic analysis, both experimentally and theoretically. Scale invariant determination of *G* is viable in principle, therefore turbulent flow, necessarily scale invariant [9], is possible both in the interior and two-dimensionally in the exterior layer.

It has been argued recently that *G* is computable from observations in a scale invariant medium, and shown to work [9]:
(1)$$G\equiv -\frac{K\left(q\right)}{q}$$
where *K*(*q*)/*q* is a scaling quantity related to partition function *f*, Boltzmann constant *k* and temperature *T* by:
(2)$$T\equiv \frac{1}{kq}$$
(3)f≡exp{−K(q)}


The relationship of *K_q_* to the Hurst exponent *H* is given in [9] as
(4)$$H=H_{q}+K(q)/q$$
where
(5)$$H_{q}=\frac{\zeta \left(q\right)}{q}$$
and *ζ*(*q*) is the linear slope of a log–log plot of the first order structure function of the fluctuations of the observed variable versus a lag parameter covering the range of the variable. By applying scale invariance to the bulk medium, and separately to the microdroplet, the Δ*G* between the two may be obtained by difference. Application of high-resolution experimental and imaging techniques would then enable a comparison with the values obtained by quantum statistical methods applied to the reactant molecules. It is noted for example that the Hurst exponent *H* has been obtained for essential genes in bacteria [22].

A comparison of scale invariant non-equilibrium thermodynamic quantities with their molecular equilibrium equivalents for a particular reaction will involve consideration of the van’t Hoff equation in the molecular equilibrium case:**K**_eq_ = exp(−Δ**G**/**RT** = exp(−Δ**H**/**RT**)exp(Δ**S**/**RT**)(6)
where
(7)G=H−TS
Here, the bold non-italic **G**, **R**, **T**, **H**, and **S** represent respectively equilibrium values of Gibbs energy, gas constant, temperature, enthalpy and entropy. **K_eq_** is equal to the ratio of the reaction rate coefficients of the forward and back reactions of the overall reaction under consideration, and allows consideration of the separate enthalpic and entropic contributions. The scale invariant equivalents are given in Table 1 of reference [9]. While equilibrium exercises have been carried out for microdroplets [3,4,5,6,7,8], none have been possible yet for scale invariant cases because of the lack of suitable observations.

## 3. Results

Microdroplets are common in a wide variety of processes and can occur in fluid media having an extensive range of scales, ranging from laboratory reactors to planetary atmospheres; aerosols in Earth’s atmosphere have long been regarded as the single largest source of uncertainty in assessing the radiative balance in climate change [23]. They have been proposed as venues for prebiotic chemistry [20] and occur frequently in electrospray mass spectrometry analysis in biochemistry [1,2,7,8].

A result of the above considerations is the possibility of summing the chemical potentials *μ* of the *i* individual molecules—their Gibbs free energy—in both the surface film and interior to compute *G*_2D_ and *G*_3D_ for the microdroplet:
(8)$$\sum_{i}^{2D}\;\mu =G_{\mathrm{film}}$$
(9)$$\sum_{i}^{3D}\;\mu =G_{\mathrm{interior}}$$
The results of such a procedure, for what is in many circumstances a non-equilibrium system, could be compared with the equilibrium results computable via quantum statistical mechanics. Recall that the transport between interface and interior will be fluid mechanical and turbulent, enabling scale invariant calculation, via observation, of *G* and hence Δ*G*.

Ultraviolet and visible light are of the appropriate energy to break chemical bonds, and so photochemistry is a widely applied technique and is also involved in atmospheric aerosols [24]. In the case of microdroplets, incident UV/Vis fluxes will have longer photon pathlengths in the crowded interior, which will increase the energy density and hence lessen dissipation. Dissipation, whether by thermal energy released in reaction or by infrared radiation, is lessened because of the lower volume available for entropy production by expansion. Both effects will tend to make Δ*G* more favorable for reaction. Figure 1 illustrates the process for polymerization of amino acids [24]. Note that such reactions can, in principle, transform Δ*G* to enable microdroplet division [15].

## 4. Discussion and Conclusions

Investigating microdroplets experimentally is likely to be challenging, particularly in regard to the spatial resolution necessary to obtain *G* by scale invariance. The effects of charge during mass spectrometric analysis must also be dealt with, to distinguish neutral acceleration from ion–molecule effects. Imaging has shown similar effects of surface reaction, however [7,25].

Molecular dynamics (MD) calculation of the behavior of molecular populations characteristic of a microdroplet are possible; the molecules are sufficiently numerous to display thermodynamic behavior, having shown scaling behavior in smaller MD calculations [26]. The combination of MD calculations and high-resolution experimental observations seems to offer a promising way forward. The experimental observations of long-tailed mass spectra featuring polymers of polar hydrocarbons in atmospheric aerosols offers a promising starting point. It will be fascinating in future to see if these considerations are relevant to the complex biochemical kinetics happening in single-celled bacteria, which are of the same size as microdroplets and atmospheric aerosols. Where more than one reaction is concerned, Δ*G* will refer to the change between the exterior medium and the microdroplet as a whole. Determining the chemical potential of biopolymers—the Gibbs energy of individual substances—by scale invariance [9] raises some interesting possibilities via summation for the determination of overall Δ*G*, both in the 2D exterior film or membrane and the 3D interior (see Equations (8) and (9)). It is noted that the need for progress in understanding fluid mechanics and free energy in biological systems has been emphasized recently [27,28].

Finally, comparison between *G* and Δ*G* computed using scale invariant Equations (1)–(5) and (8)–(9) with the equilibrium statistical thermodynamic Equations (6) and (7) will have potential for diagnosing the state of a microdroplet, particularly for determining whether the system is in a stationary state or is at true equilibrium. Techniques for the analysis may be found in references [9], [26], [29] and [30].

## Figures and Tables

**Figure 1 entropy-21-01044-f001:**
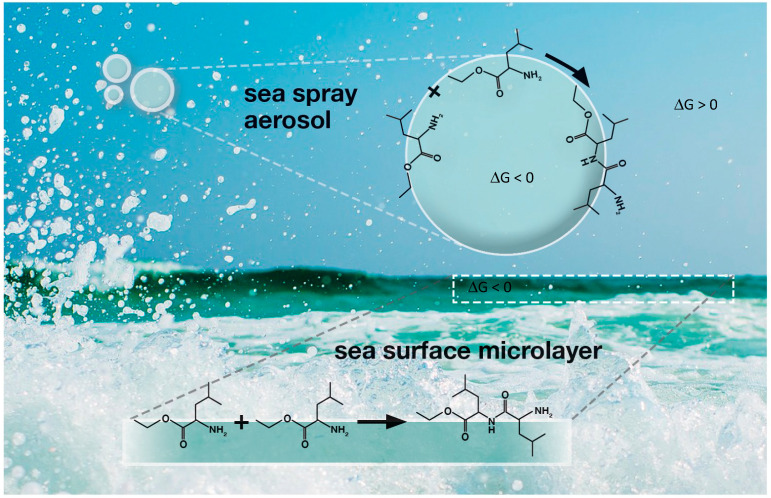
Showing the polymerization of an amino acid to an oligomer on the surface of a marine microdroplet [24] with the associated Gibbs free energy changes (Δ*G*) régimes.

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
