# Peer review of "Gibbs Free Energy and Reaction Rate Acceleration in and on Microdroplets"

_entropy, 2019, doi:10.3390/e21111044_

Round 1

Reviewer 1 Report

This manuscript addresses an important issue vis. the change of free energy ΔG of reactions when going from for example the gas pahse to that in micro-droplets or films. It is well written and well argued.

The methodology developed is innovative. An approach is presented, which explains how the change in Gibbs free energy ΔG for physico-chemical reactions and Gibbs free energy G at the surface layer and within the interior of a droplet could be calculated using or applying scale invariance.

The author argues that these ΔG changes are “relatable to the crowding enforced by microdroplets and to scale invariance.”  See Line 106 "By applying scale invariance to the bulk medium, and separately to the microdroplet, the ΔG between the two may be obtained by difference."

For this approach to be used for reactions on microdroplets or films, it is necessary to calculate the G values. It is noted that molecular dynamic calculations are one way to do this. These are probably beyond the scope of this study.

Comments

The speed of a reaction between molecules A and B

                                       A+ B à C + D

is generally considered to depend on the Gibbs free energy difference, ΔG, (sometimes simplified to the enthalpy difference ΔH i.e. ignoring changes in entropy ΔS, between the transition state and the reactants.

Consequently I consider that the reason for the increase of speed of the reaction is the lowering of ΔG (= the difference between the sum of G for the reactants A and B, and that of the transition state AB*) between that in

the gas phase, where it is presumably large and thus a barrier to reaction, and the quasi-liquid or liquid surface or interior of a microdroplet.

This may be equivalent to what the author has written but it would be valuable to clarify this point for the more general reader, who is not so familiar with scale invariance. It may well be that the entropy differences, ΔS,  for the different conditions dominates over the changes in enthalpy ΔH in lowering the overall  Gibbs free energy difference, ΔG.

For equilibrium reactions between A and b producing C and D in accordance with the van`t Hoff equation and conventionally

K = exp(-ΔG/RT) = exp(-ΔH/RT)exp(ΔS/R)

Thus changes in G for the reactants and products between the gas phase and the quasi-liquid or liquid surface or interior of a microdroplet.

I recommend that the author takes a simple or idealized example from atmospheric chemistry and calculate the change in ΔG for a selected simple case i.e. simpler than the process for polymerization of amino acids, which is currently the example used.

This would enable him to demonstrate the acceleration of reaction calculated in the way the manner he proposes.

Minor comment

Line 93 an r is missing it should  read "related" not elated.

Conclusion

After consideration of my suggestion to improve the manuscript by the author, I recommend publication.

Author Response

This manuscript addresses an important issue vis. the change of free energy ΔG of reactions when going from for example the gas pahse to that in micro-droplets or films. It is well written and well argued.

Author's responses: *Thank you, comment appreciated.

...... Thus changes in G for the reactants and products between the gas phase and the quasi-liquid or liquid surface or interior of a microdroplet.

Author's responses: These remarks are of course correct and relevant. I have added in text and equations to respond.

I recommend that the author takes a simple or idealized example from atmospheric chemistry and calculate the change in ΔG for a selected simple case i.e. simpler than the process for polymerization of amino acids, which is currently the example used.

Author's responses: Much as I would like to be able to do this, it’s not possible because the method is dependent upon the availability of adequate observations and experimental results; currently none are. I have added some references that deal with simpler, atmospheric molecules, albeit without the resolution required for scale invariant analysis. The aim of the paper is to present the framework for the approach.

Minor comment

Line 93 an r is missing it should  read "related" not elated.

Author's responses: done.

Reviewer 2 Report

This paper is focussed on the potential connection between scale invariance and the acceleration of reaction rates inside microdroplets. This is a novel idea, and while investigations into reactions both within and at the surface of microdroplets are of widespread interest right now, there is no solid argument presented in this paper that connects scale invariant phenomena with chemical reactions within microdroplets. Indeed, the paper as written is not clearly structured, and there are no arguments or calculations presented in the “Results” section to support the initial hypothesis.

The formulation of thermodynamic quantities based on observed scale invariances in the atmosphere presented in reference 9 (Tuck, J. Phys. Chem. 2017) is promising, but I believe it may be misplaced when considering chemical processes within a microdroplet. In general, scale invariant systems have no characteristic scale, but microdroplets very much do have a characteristic scale – the molecular scale – and it is this scale that is driving at least part of the enhancement of reaction rates being reported in the literature. Reactions in microdroplets differ from reactions in the bulk in part because the extremely high surface to volume ratio of the droplets which gives rise to a number of processes at the air-liquid interface that plausibly lead to enhancement in reaction rates (effects from surface charge, molecular orientation, and contact ion-pairing are unique to microdroplets compared to the bulk-phase and these are not scale invariant processes).  The author’s arguments for scale invariance applying to such systems should be clearly stated along with a careful explanation of which structural or dynamical properties need to be assumed scale invariant for this line of reasoning to hold.

While I recommend that this MS be rejected, I would with encourage the author to resubmit a restructured paper in which the ideas are better laid out; ideally these would then be supported by theoretical or experimental findings.

Author Response

Authors’ responses:

I appreciate the effort this reviewer has made. The Alder & Wainwright (1970) paper in reference [9] combined with reference [26], Kadau et al.,(2008) in the revised manuscript makes it clear that scale invariance has an inner and an outer scale. The outer scale is that of the microdroplet, just as the Earth’s circumference is the outer scale for the atmosphere. The inner scale, as stated in the paper, will be a few mean free paths, allowing a scale invariant analysis in principle. Fluid mechanics applies within a microdroplet, as follows from the above references.I would very much like to use experimental results to demonstrate the approach, but cannot at present because adequate experimental data do not exist. I still wrote the paper for Entropy because its website encourages the submission of new ideas and approaches, which all the reviews say is the case. The revised manuscript has coniderations of how G, H, T, and S would appl via the van’t Hoff relation in equilibrium statistical thermodynamics. While reactions are taking place in a flow however, the system is not at equilibrium. The point of the approach in this paper is to offer an analysis with which the equilibrium result could be applied. I note that Reviewer 1 has a more positive view than Reviewer 2.

Reviewer 3 Report

This paper shows the significance of Gibbs free energy changes and the implications for chemistry.

The introduction could be improved, adding more info and references mainly when talking about atmospheric chemistry.

Results and conclusions sections have to be separated, since it is labelled as results but the author put the conclusions there. It is complicated to identify where the conclusions start. There are some comments (e.g. line 145-146) that sounds more as conclusions than as results. Therefore, it is needed the addition of a Conclusions section perfectly defined and completed. 

Author Response

This paper shows the significance of Gibbs free energy changes and the implications for chemistry.

The introduction could be improved, adding more info and references mainly when talking about atmospheric chemistry.

Author’ responses: The Introduction has been amended, and includes references that are appropriate to atmospheric chemistry.

Results and conclusions sections have to be separated, since it is labelled as results but the author put the conclusions there. It is complicated to identify where the conclusions start. There are some comments (e.g. line 145-146) that sounds more as conclusions than as results. Therefore, it is needed the addition of a Conclusions section perfectly defined and completed.

Author’ responses: The results have been separated from a new section, 4, Discussion and conclusions has been added.

Round 2

Reviewer 2 Report

Although I acknowledge the author's efforts in addressing some of my concerns, the fundamental issue to me remains that there is no real substance to this paper. It introduces the concept of using scale-invariance to calculate G separately for surface and bulk in a microdroplet, and to then take the difference between these 2. However, there is no real pathway shown for how this might be done (other than some suggestion of various potential methods), nor is the major issue of how to apply this to complex, multicomponent systems even raised.

If the editors are happy with a 'blue sky ideas' paper, this is certainly a good one.

Author Response

This has always been an explicitly "blue sky" paper. I realise that it may not gel well with, for example, typical papers in current physical chemistry journals - long ago, I was trained as a physical chemist myself before converting to geophysics. One reason why not is the relative novelty of scale invariant thermodynamics, combined with the new idea of applying it to the recently realised importance of reactions in and on microdroplets. I have added text that I hope will at least partially assuage the reservations of reviewer #2. The challenge posed by the proposed research strategy should interest experimental physical chemists and chemical physicists.